# Branch Duct IPMN-Associated Acute Pancreatitis in a Large Single-Center Cohort Study

**DOI:** 10.3390/diagnostics15131676

**Published:** 2025-07-01

**Authors:** Nicolò de Pretis, Luigi Martinelli, Antonio Amodio, Federico Caldart, Salvatore Crucillà, Maria Sole Battan, Alberto Zorzi, Stefano Francesco Crinò, Maria Cristina Conti Bellocchi, Laura Bernardoni, Giulia De Marchi, Pietro Campagnola, Roberto Salvia, Armando Gabbrielli, Alessandro Marcon, Luca Frulloni

**Affiliations:** 1Gastroenterology Unit, Department of Medicine, University of Verona, P.le L.A. Scuro 10, 37134 Verona, Italy; 2Department of Diagnostics and Public Health, University of Verona, 37134 Verona, Italy; 3Department of Surgery, University of Verona, 37134 Verona, Italy; 4Gastroenterology Unit, University of Trento, 38122 Trento, Italy

**Keywords:** IPMN, acute pancreatitis, pancreatitis, cancer, recurrent pancreatitis, relapse

## Abstract

**Background/Objectives**: IPMNs are a possible cause of acute pancreatitis (AP). IPMN-associated-AP is considered a relative indication for surgery for the prevention of recurrent attacks of pancreatitis and for a hypothesized increased cancer risk. The literature is based on surgical series, and no data on the clinical features of AP associated with IPMNs and on the risk of recurrent pancreatitis and pancreatic cancer are available. This study aimed to describe the clinical/radiological features of BD-IPMN-associated AP. Moreover, BD-IPMN-associated risk factors for AP recurrence and risk of pancreatic cancer were investigated. **Methods**: Patients with AP associated with branch-duct IPMN (BD-IPMN) without “worrisome- features” and “high-risk-stigmata” evaluated in a Gastroenterology Unit (University of Verona) between 1 January 2012 and 31 December 2022 were retrospectively analyzed. Cox proportional hazard models were used to analyze the time to recurrence after the first occurrence of AP. **Results**: One hundred and thirty-five patients were included, with a mean age of 55.8 ± 12.5 years. Necrosis was diagnosed in 15 patients (11.1%) and 1 patient (0.7%) was admitted to the ICU. One hundred and two (75.6%) patients had recurrent pancreatitis. The median size of the largest BD-IPMN was 8 mm (Q1–Q3: 5–12). Eighteen patients (13.3%) developed main pancreatic duct dilation ≥ 5 mm. No patients developed dilation of the main pancreatic duct ≥ 10 mm, mural nodules, thickened cystic walls, or jaundice. In the unadjusted analysis, no BD-IPMN-related features were associated with an increased risk of recurrent pancreatitis. None of the patients developed pancreatic cancer. **Conclusions**: BD-IPMN-associated AP appears to have a benign clinical course. Cystic features related to increased risk of recurrence were not identified. The risk of cancer appears extremely low.

## 1. Introduction

Detection of branch-duct intraductal papillary mucinous neoplasms (BD-IPMNs) has increased over time. BD-IPMNs are frequently discovered in asymptomatic individuals, and their management is tailored based on the presence of symptoms and radiological appearance. If high-risk stigmata, such as obstructive jaundice or an enhanced solid component, are detected, fit patients should undergo surgical resection. In cases of worrisome features (cyst size ≥ 3 cm, thickened enhanced walls, non-enhanced mural nodules, and acute pancreatitis) [1,2,3], strict surveillance or surgical resection might be considered. If BD-IPMNs presents radiological features suggesting a low risk of pancreatic cancer, surveillance is suggested, considering the low risk of malignant transformation [4,5].

All IPMN subtypes (MD-IPMN, mixed-type IPMN and BD-IPMN) are potential causes of acute pancreatitis (AP) [6]. The hypothesis is that mucin produced by the cyst might obstruct the main pancreatic duct, leading to transitory ductal hypertension and, subsequently, AP [7,8,9,10]. This hypothesis is also supported by the reported clinical benefit of pancreatic sphincterotomy in terms of reducing relapsing attacks of AP in patients with IPMN-associated AP [9,11,12].

However, considering that reliable diagnostic tests are lacking, the diagnosis of IPMN-associated AP is achieved by the exclusion of other possible causes of pancreatitis in patients with BD-IPMN detection on high-resolution imaging performed during the first attack of AP. If the cystic lesion develops after the first attack of AP, it might be considered a consequence of pancreatic inflammation, such as pseudocysts or pancreatic fluid collections. Some authors have suggested that IPMN-associated AP is related to an increased risk of malignant transformation [13]. For this reason, and to reduce the risk of relapsing episodes of AP, surgical resection has been proposed in these patients [3,14,15,16], especially in cases of recurrent pancreatitis (>1 episode of AP during follow-up). Some studies have suggested that surgical resection may reduce the risk of further attacks of AP [17]. As previously reported, in patients unfit for surgery or as a bridge to pancreatic surgery, pancreatic sphincterotomy has also been proposed to facilitate mucin outflow and reduce the risk of recurrent attacks of pancreatitis [9]. However, in contrast with surgical resection, endoscopic treatment does not influence the risk of malignant transformation of BD-IPMNs. Therefore, a better understanding of the natural history, clinical features, and cancer risk in patients with BD-IPMN-associated AP is necessary to introduce conservative or minimally invasive strategies in clinical practice.

The available data on IPMN-associated AP are mainly based on surgical series, with heterogeneous data including patients with different IPMN subtypes. No studies have investigated AP specifically in patients with BD-IPMNs, who are at a lower risk of cancer development. Moreover, previously published studies did not investigate the clinical features of BD-IPMN-associated AP in terms of severity, and no data are available on the risk of cancer in non-surgically resected patients.

The main aim of this study was to describe the clinical and radiological features of BD-IPMN-associated AP. Moreover, we investigated the BD-IPMN-associated risk factors for AP recurrence. Finally, we evaluated the risk of pancreatic cancer in this population.

## 2. Methods

All patients with AP associated with BD-IPMN evaluated at the Gastroenterology Unit of the University of Verona between 1 January 2012 and 31 December 2022 were identified. This study was approved by the local ethics committee (Ethics Committee of the Province of Verona and Rovigo, protocol number 1271CESC, April 26th, 2017). Informed consent was obtained from all subjects involved in the study in accordance with the Declaration of Helsinki.

The inclusion criteria were as follows:-At least one episode of AP based on the Atlanta classification, which requires the presence of two of the following three criteria: typical abdominal pain (pain consistent with acute pancreatitis (acute onset of a persistent, severe, epigastric pain often radiating to the back), serum amylase and/or lipase > 3 times the upper normal limit, and high-definition imaging (contrast-enhanced CT scan and/or contrast-enhanced MRI scan) results suggestive of pancreatitis.-At least one available set of high-resolution imaging results (contrast-enhanced CT scan and/or contrast-enhanced MRI scan) obtained within 7 days of the clinical onset of the first AP attack, showing the presence of a pancreatic cyst suggestive of BD-IPMN.-At least a second round of high-resolution imaging performed > 3 months later.The exclusion criteria were as follows:-Signs of chronic pancreatitis observed using the first high-resolution imaging technique (relevant signs include pancreatic atrophy, dilation of the main pancreatic duct ≥ 5 mm, and ductal or parenchymal calcifications).-Signs of MD-IPMN (mixed type) on the first high-resolution imaging technique (main pancreatic duct ≥ 5 mm).-The presence of symptoms or cystic features observed via the first high-resolution imaging technique, suggesting upfront surgical resection, such as mural nodules, thickened cystic walls, jaundice, or solid neoplastic tissue near the cystic lesion.-History of previous pancreatic surgery or pancreatic sphincterotomy.-Less than one year of follow-up.-Other causes of acute pancreatitis are defined as follows:
(a)Biliary pancreatitis: bile duct stones are detected on imaging or a transient elevation of transaminase is detected at the clinical onset of acute pancreatitis.(b)Alcohol-related pancreatitis: 20 g/day of chronic alcohol consumption and exclusion of other causes of acute pancreatitis.(c)Hypertriglyceridemic pancreatitis: serum triglycerides at clinical onset > 500 mg/dL.(d)Genetic pancreatitis: detection of gene mutations (CFTR, SPINK-1, PRSS-1), which are conventionally investigated if the first attack of acute pancreatitis occurred before the age of 40 years.(e)Other causes of pancreatitis: in cases of tumor or significant ductal or parenchymal abnormalities on imaging, clinical, demographic, radiological, and pathological data were analyzed. Diabetes was defined according to the American Diabetes Association (ADA) criteria [18]. The presence of pancreatic necrosis was confirmed using high-resolution imaging.



The radiological features of BD-IPMN were recorded via high-resolution imaging performed during the first attack of AP and the last available high-resolution imaging session. On the initial high-resolution images, pancreatic cysts were considered as BD-IPMNs in case of (a) a cystic lesion > 5 mm in diameter communicating with the MD; (b) multifocal cysts, at least one of which was clearly communicating with the main pancreatic duct; and (c) cystic lesions without clear communication with the main pancreatic duct that were clinically managed as BD-IPMNs [19]. In patients with ≥1 BD-IPMNs, the radiological characteristics of the largest cyst were recorded. According to the revised International Consensus Fukuoka guidelines and the international evidence-based Kyoto guidelines, “high risk stigmata” (indicative of the need for surgery) included jaundice, enhancing mural nodules ≥ 5 mm, and MPD ≥ 10 mm. “Worrisome features” (perceived as a relative indicator for surgery) included enhancing mural nodules < 5 mm, a thickened and enhancing cyst wall, MPD dilation 5–9 mm, an abrupt change in MPD caliber, and a cyst growth rate > 5 mm/2 years [5,14,15]. Cancer was considered if it was detected via pathology in patients treated with surgery or in cases of typical radiological features on high-resolution imaging.

Our research followed the Strengthening the Reporting of Observational Studies in Epidemiology (STROBE) statement of observational studies. The authors declare that they have no conflicts of interest.

### Statistical Analysis

Patient and cyst characteristics at baseline were described through contingency tables, showing number and proportions (%) for qualitative variables, and means with standard deviations or medians with inter-quartile ranges for symmetrical quantitative variables, respectively. A time-to-event analysis was performed by applying both univariable and multivariable proportional hazard Cox models. The first recurrence was considered as a failure event, while the months between the first AP and the first recurrence were considered as the time scale. The proportionality assumption was checked by graphic methods: it was verified whether −ln[−ln[survival]] curves for each category of risk factors were parallel when plotted vs. ln [analysis time].

Statistical analyses were performed using STATA software, release 18 (StataCorp LLC, College Station, TX, USA).

## 3. Results

### 3.1. Clinical and Radiological Features of the Population

One hundred and thirty-five patients met the inclusion criteria, with a mean follow-up of 5.2 ± 6.0 years. Seventy-four were males (54.8%) and sixty were females (45.2%), with a mean age at the first episode of AP of 55.8 ± 12.5 years. Diabetes was present at clinical onset in 16 patients (11.8%), whereas smoking and alcohol consumption (<20 g/day) were reported in 64 (47.4%) and 41 (30.4%) patients, respectively.

Pancreatic necrosis was diagnosed via imaging in 15 patients (11.1%) and only 1 patient (0.7%) was admitted to the Intensive Care Unit for the clinical management of AP. None died. One hundred and two (75.6%) of the included patients had recurrent pancreatitis (more than one episode of AP with no other identifiable causes), with a median recurrence time of 26 months (Q1–Q3: 10–65), while thirty-three (24.4%) had a single episode of AP. Thirty-six patients (26.7%) were evaluated at our center for their first episode of AP. Among this cohort, the risk of recurrence was 5.5% at 1 year and 11.0% at 3 years.

Table 1 summarizes the main clinical feature of the study population.

### 3.2. Risk of Recurrent Pancreatitis

None of the patients’ clinical characteristics were associated with an increased risk of recurrence in the unadjusted and adjusted Cox regression analyses (Table 2.). Among the 135 patients, 73 (54.1%) had a single BD-IPMN, whereas 62 (45.9%) had multifocal BD-IPMNs on the high-resolution imaging performed for the first episode of AP. No significant differences in the risk of recurrence were detected between patients with 1 or >1 BD-IPMNs in the unadjusted and adjusted Cox models (Table 2).

The median size of the largest BD-IPMNs in the imaging technique performed during the first episode of AP was 8 mm (Q1–Q3: 5–12) and only in 6 patients (4.5%) the largest BD-IPMNs was > 30 mm. None of the patients had mural nodules, thickened cystic walls, or a dilated main pancreatic duct of ≥5 mm. Among the 33 patients who had experienced a single episode of acute pancreatitis, 32 had the largest BD-IPMN ≤ 30 mm (96.7%), while among 102 patients with recurrent pancreatitis 97 had the largest BD-IPMN ≤ 30 mm (95.1%) (*p* > 0.99). No significant relationship between cyst size and the risk of recurrence was detected in the unadjusted and adjusted Cox models. (Table 2 and Figure 1).

During the high-resolution imaging performed at the first AP, BD-IPMNs were limited to the head/uncinate process in 65 patients (48.1%) and at body/tail in 36 patients (26.6%). In 34 patients (25.2%), BD-IPMNs were diffusely detected throughout the entire pancreas. In the unadjusted and adjusted Cox analyses, the location of the BD-IPMN was not related to the risk of recurrence (Table 2 and Figure 2).

### 3.3. Follow-Up

The last available imaging diagnostics, performed after a median time of 24 months (Q1–Q3: 12–48), revealed that the largest BD-IPMNs significantly increased in median size (8 mm Q1–Q3: 5–12 vs. 10 mm Q1–Q3: 6–16 *p* = 0.01). Moreover, the number of patients with the largest BD-IPMNs > 30 mm was 6 (4.5%) at the first episode of acute pancreatitis and 13 (9.6%) according to the last set of images obtained. Eighteen patients (13.3%) developed main pancreatic duct dilation of ≥5 mm. None of the patients developed dilation of the main pancreatic duct of ≥10 mm, mural nodules, thickened cystic walls, or jaundice. Globally, “worrisome features” were identified in 6 patients (4.5%) at the first episode of AP (only for size ≥ 30 mm) and in 23 patients (17.0%) at the end of follow-up. This difference was statistically significant (*p* = 0.001).

None of the patients developed “high risk stigmata,” pancreatic cancer or signs of chronic pancreatitis during follow-up. We did not find significant differences in pancreatitis recurrence risk in patients developing dilation of ≥ 5 mm of the main pancreatic duct compared with patients with stable main pancreatic duct size. (Table 2.) Surgical resection was performed for main pancreatic dilation or to reduce pancreatitis recurrence in 10 patients (7.4%), all with at least two episodes of AP, while 7 (70%) underwent surgery after more than two episodes. Eight patients underwent pylorus-preserving duodeno-pancreatectomy and two underwent distal pancreatectomy after a mean time of 56.4 ± 29.9 months from the first AP and after a median of three (IQR 3) episodes of acute pancreatitis. Histology revealed BD-IPMNs with low-grade dysplasia in seven patients (70%) and BD-IPMNs with high-grade dysplasia in three patients (30%). No pancreatic cancer diagnoses were made based on surgical specimens obtained from this population. No predictive factors for HGD were identified. During the follow-up, no other causes of pancreatitis were identified in the study population. Figure 3 shows BD-IPMN on MRI, EUS, and pathology.

## 4. Discussion

BD-IPMNs are a critical topic in clinical practice, and conservative management is being proposed increasingly often. Surgical resection is reserved for patients with a high risk of harboring or developing pancreatic cancer. AP in patients with IPMNs has been proposed as an indication for surgery as it is potentially associated with an increased risk of pancreatic cancer, and such a surgery may reduce the recurrence of pancreatitis [1].

Case series and cohort studies on IPMN and AP have been published, mainly based on patients who underwent surgical resection. These studies mostly reported heterogeneous populations, with limited sample sizes, including MD-IPMN, mixed-type IPMN, and BD-IPMN. Moreover, other potential causes of AP have not been clearly excluded. Moreover, the European Guidelines report that the level of evidence of the association between pancreatitis and cancer in patients with IPMN is low and is based predominantly on surgical series [3].

In the present study, only patients with BD-IPMN-associated AP were included, without radiological evidence of pancreatic cancer, “high-risk stigmata,” or “worrisome features” (with the exception of size) at the first episode of pancreatitis. Other potential pancreatitis causes were carefully excluded. Moreover, the large sample size (135 patients) and median radiological follow-up are relevant for the characterization of this population. We showed that BD-IPMN-associated AP has a generally favorable clinical course with low frequency of pancreatic necrosis (about 11%), ICU admission < 1%, and 0% of deaths, which should always be considered when invasive procedure or surgery are proposed to reduce the risk of recurrence. To our knowledge, no previously published data on necrosis and risk of ICU admission are available in this clinical setting.

Seventy-five percent of our patients had recurrent pancreatitis, and this high proportion was probably related to referral bias. The Gastroenterology Unit of the University of Verona is considered a tertiary center for pancreatic diseases and patients have a higher probability of being referred due to recurrent attacks rather than after a first episode of AP.

Interestingly, the number of IPMNs did not influence the risk or time of recurrence, similar to the location and size of the cyst. Moreover, the median size of the largest BD-IPMN was only 8 mm, with no evidence of a higher risk of recurrence in patients with larger cysts, even those >30 mm. Despite the increase in the size of BD-IPMN during follow-up, none of the patients developed “high-risk stigmata” or cancer. Sahora and colleagues reported on a surgical series of 30 patients with BD-IPMN-associated pancreatitis.

Of these patients, 15 underwent surgical resection, while 1 (3.3%) patient had cancer at pathology [20]. Another study from Japan reported similar incidence of acute pancreatitis in 12 malignant and 32 non-malignant BD-IPMNs in a surgical retrospective series; 18.7% and 16.7%, respectively [21]. Moreover, among the eight surgically treated patients with BD-IPMNs and associated pancreatitis, two presented signs of cancer at pathology (25%). Of seven patients resected in a referral US center for BD-IPMNs with associated acute pancreatitis, two (28.6%) had high-grade dysplasia/invasive cancer [22]. Finally, a study from Korea reported a cancer risk of up to 48% in patients with IPMNs and associated pancreatitis [13] and a recently published study from the US reported 15 patients with resected BD-IPMNs with previous history of recurrent pancreatitis. Only four of them had worrisome features/high-risk stigmata but five (33%) showed signs of malignancy on surgical pathology [16]. The proportion of cancer reported in these studies was higher than that reported in the present study, which included a significantly larger sample size, without evidence of cancer on imaging and pathology. This difference is likely related to the different clinical settings from which the included patients originated. Most of the literature is based on surgical series [17,23,24,25], without excluding IPMN with involvement of the main pancreatic duct. In contrast, the present study included only patients with BD-IPMN, without worrisome features or high-risk stigmata at the time of the first episode of AP (except for size, which is less often considered as an indication for surgery).

Main pancreatic duct dilation was observed during follow-up in 13% of cases, with consequent reclassification of BD-IPMNs into mixed-type IPMNs. None of the patients developed high-risk stigmata. Finally, among 10 surgically resected patients (7%), 3 had high-grade dysplasia and none had cancer at pathology. These data suggest that clinical and radiological surveillance might be an acceptable option even in patients with BD-IPMN-associated AP, supporting more conservative management as proposed for BD-IPMN at low oncological risk [26,27,28,29]. Therefore, endoscopic pancreatic sphincterotomy might be considered not only for inoperable patients and patients waiting for surgery [9] but also for fit patients with low-risk BD-IPMNs. Further prospective studies are required to confirm these findings.

This study is the first to focus exclusively on BD-IPMN-associated AP. It reports the largest sample size ever published on this topic, with a reasonable follow-up time. However, this study has some limitations. The first criticism, which similarly applies to all of the published literature on this topic, is the lack of definitive and reliable diagnostic tests to establish with certainty that the cause of AP is BD-IPMN. However, the identification of IPMNs as a cause of AP is only based on the exclusion of other possible causes of AP, and during follow-up, no other causes of pancreatitis were identified.

Additionally, the study was retrospective with possible referral bias, considering that our unit is a tertiary center for pancreatic diseases. However, despite the high proportion of patients with recurrent pancreatitis, possibly indicative of more aggressive disease, no patients developed cancer, suggesting that even in patients with BD-IPMN-associated recurrent pancreatitis without high-risk stigmata or worrisome features, the risk of cancer appears low. Moreover, we did not identify any BD-IPMN-associated features related to a higher risk of pancreatitis recurrence. Finally, although this is the largest series in the literature on this topic, larger studies with longer follow-up are needed to confirm these results.

In conclusion, BD-IPMN-associated AP has a favorable clinical course with a negligible risk of Intensive Care Unit admission and death. BD-IPMNs are frequently small in this clinical setting, and no radiological features related to BD-IPMN are associated with an increased risk of recurrence. BD-IPMN tends to grow slowly over time in this clinical setting and a small proportion may develop main pancreatic duct dilation during follow-up. None developed “high risk stigmata” or pancreatic cancer during follow-up. Radiological surveillance is likely feasible in these patients and surgery should be reserved for patients with radiological features suspected of degeneration. To reduce the risk of AP relapses, less invasive treatment such as pancreatic sphincterotomy might be preferred to surgery if radiological features suggestive or suspected of malignant transformation are lacking. Surgery may be considered to decrease recurrent attacks of AP after the failure of endoscopic treatment.

## Figures and Tables

**Figure 1 diagnostics-15-01676-f001:**
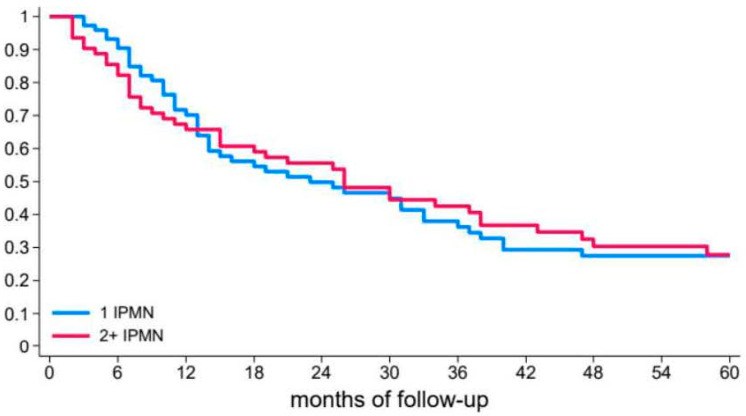
Risk of relapse after the first episode of acute pancreatitis based on number of BD-IPMNs; 1 IPMN: a single cyst (blue line); 2 + IPMN: more than 1 cyst (red line).

**Figure 2 diagnostics-15-01676-f002:**
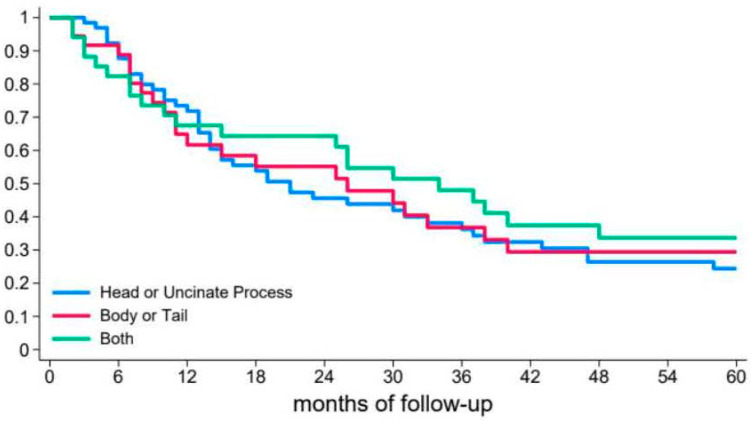
Risk of relapse after the first episode of acute pancreatitis according to the location of BD-IPMN at the head or uncinate process (blue line), at body or tail (red line) and in diffuse forms (green line); diffuse is presence of BD-IPMN on head/uncinate process and on body/tail.

**Figure 3 diagnostics-15-01676-f003:**
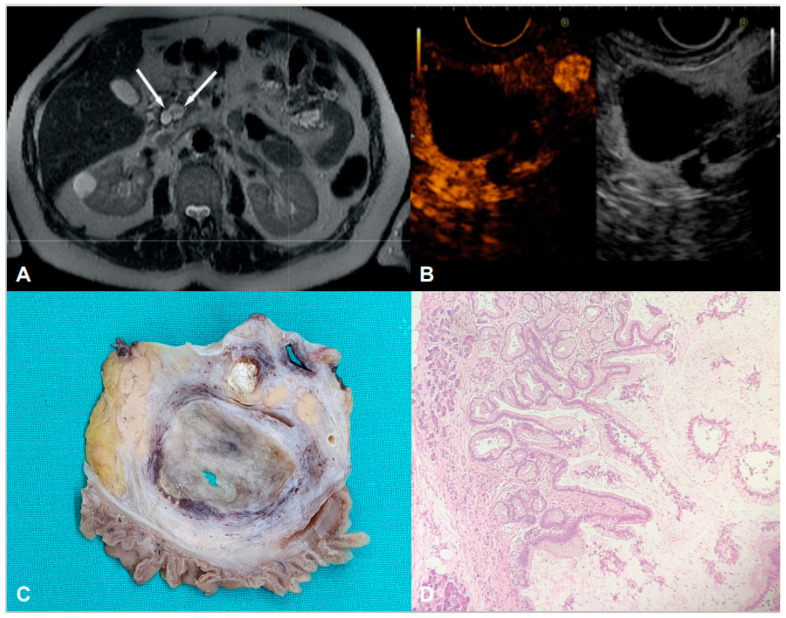
BD-IPMN in a patient with BD-IPMN-associated acute pancreatitis on MRI (**A**), EUS (**B**), and macroscopic (**C**) and microscopic (**D**) pathology images. Pathology was conclusive for the gastric subtype with low-grade dysplasia.

**Table 1 diagnostics-15-01676-t001:** Main clinical features of the study population. Nr. is number, SD is standard deviation, and IQR is inter-quartile range. Age is referred to as the first episode of acute pancreatitis in patients with recurrent pancreatitis.

Patients nr. (%)	135 (100)
Males nr. (%)	74 (54.8)
Age (years) mean (SD)	55.8 (12.5)
Diabetes nr. (%)	16 (11.8)
Alcohol nr. (%)	41 (30.4)
Smoke nr. (%)	64 (47.4)
Recurrent pancreatitis nr. (%)	102 (76.6)
Recurrence time (months) median (p25–p75)	15 (7–37)
Pancreatic necrosis nr. (%)	15 (11.1)
Intensive Care Admission nr. (%)	1 (0.7)
Follow-up (years) median (p25–p75)	5.2 (3.9–6.0)

**Table 2 diagnostics-15-01676-t002:** Cox model hazard ratio results (patients = 135) evaluating the risk of having at least one acute pancreatitis recurrence. CI: confidence interval is interval of confidence. Alcohol consumption of >0 and ≤20 g/day was recorded. MPD: is main pancreatic duct. The MPD was considered to be dilated if, during follow-up, the diameter was ≥ 5 mm (patients with MPD ≥ 5 mm on the high-resolution imaging performed at the first episode of pancreatitis were not included in the study). There were no missing data for all the variables included in the analysis.

	Characteristics	Unadjusted Hazard Ratio (95% CI)	*p* Value	Adjusted Hazard Ratio (95% CI)	*p* Value
**Characteristics of Patients**	**Age at 1st AP**	1.00 (0.99–1.02)	0.830	1.00 (0.99–1.02)	0.650
	**Sex**				
	*Female*	1#	0.396	1#	0.539
	*Male*	0.84 (0.56–1.26)		0.86 (0.54–1.38)	
	**Smoking**				
	*No*	1#	0.626	1#	0.846
	*Yes*	0.91 (0.61–1.34)		1.04 (0.68–1.61)	
	**Alcohol**				
	*No*	1#	0.214	1#	0.407
	*Yes*	0.76 (0.50–1.17)		0.80 (0.47–1.35)	
	**Diabetes**				
	*No*	1#	0.722	1#	0.052
	*Yes*	0.50 (0.26–0.95)		0.50 (0.25–1.01)	
	**Necrosis**				
	*No*	1#	0.952	1#	0.899
	*Yes*	0.97 (0.54–1.74)		0.96 (0.50–1.86)	
**Characteristics of BD–IPMN**	**Number of cysts**				
	*1*	1#	0.242	1#	0.252
	*>1*	1.09 (0.73–1.61)		1.31 (0.82–2.10)	
	**Cyst location**				
	*Head/uncinate*	1#	0.631	1#	0.310
	*Body/tail*	0.86 (0.54–1.39)		1.02 (0.61–1.70)	
	*Diffuse*	0.80 (0.49–1.30)		0.65 (0.36–1.18)	
	**Cyst size**	1.02 (1.00–1.04)	0.099	1.01 (0.99–1.04)	0.218

## Data Availability

The data presented in this study are available on request from the corresponding author. The data are not publicly available due to privacy.

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
