# Peer review of "Branch Duct IPMN-Associated Acute Pancreatitis in a Large Single-Center Cohort Study"

_diagnostics, 2025, doi:10.3390/diagnostics15131676_

Round 1

Reviewer 1 Report

Comments and Suggestions for Authors

Dear Editor

 This is an interesting study revealing the features of BD-IPMN that could impact the subsequent clinical course. The followings are my questions. 

#1. Do you utilize EUS for IPMN assessment or only high resolution imaging ? It would be beneficial to provide illustrative images for explanations. 

#2. The figures aims to illustrate the rate of relapse after 1st pancreatitis episode. However, the Y-axis is wrong. The relapse rate should be 0 from baseline and increased gradually not from 1 to decline gradually. Please check it.  

#3. Please check the spelling and grammar, such line 238 "Seventy-five %"

#4. Please provide the reason of patients who received surgery ( Line 207 ). Is there features could predict the presence of high grade dysplasia in the 30% resected cases ? Provide radiological-histological images would be beneficial to enrich this manuscript. 

Author Response

Reviewer 1: 

This is an interesting study revealing the features of BD-IPMN that could impact the subsequent clinical course. The followings are my questions.  

Answer: Thank you very much for your comment. 

#1. Do you utilize EUS for IPMN assessment or only high resolution imaging? It would be beneficial to provide illustrative images for explanations.  

Answer: Thank you very much for this important comment. High resolution imaging is always used in patients with pancreatic cysts. EUS is reserved for patients with suspicious findings on imaging. However, patients referred by other centers have often already undergone EUS. As suggested, an image with BD-IPMN visualized at imaging, EUS, and pathology has been added. 

#2. The figures aims to illustrate the rate of relapse after 1st pancreatitis episode. However, the Y-axis is wrong. The relapse rate should be 0 from baseline and increased gradually not from 1 to decline gradually. Please check it.   

Answer: Thank you very much for this comment. We fully agree with the reviewer. The image has been modified in order to be more clear and understandable. 

#3. Please check the spelling and grammar, such line 238 "Seventy-five %" 

Answer: Than you very much for this comment. Spelling and grammar were checked. 

#4. Please provide the reason of patients who received surgery ( Line 207 ). Is there features could predict the presence of high grade dysplasia in the 30% resected cases ? Provide radiological-histological images would be beneficial to enrich this manuscript.  

Answer: Thank you very much for this important comment. Patients received surgery for pancreatic duct dilation or for pancreatitis recurrence prevention, and no predictive factors for HGD were identified. To make this point more clear, the following sentences were modified/added in the Results Section:  

  • “Surgical resection was performed for main pancreatic dilation or to reduce pancreatitis recurrence in 10 patients (7.4%), all with at least 2 episodes of AP and 7 (70%) after more than 2 episodes” 
  • “No predictive factors for HGD were identified” 

Moreover, as suggested, an image with BD-IPMN visualized at imaging, EUS, and pathology has been added. 

Reviewer 2 Report

Comments and Suggestions for Authors

The authors have effectively communicated the information in the manuscript, providing comprehensive explanations that are clear and concise. However, I have a few considerations regarding the manuscript.
1. The study's duration is only one year, and the sample size is minimal. The sample size for diabetes was merely 16, with a higher percentage of males than females. A balanced gender ratio in the sample would ensure that the findings apply to IPMN-associated pancreatitis in the research. The manuscript could benefit from more detailed explanations of acute pancreatitis, as this would enhance clarity and understanding of the topic.

Author Response

Reviewer 2: 

The authors have effectively communicated the information in the manuscript, providing comprehensive explanations that are clear and concise. However, I have a few considerations regarding the manuscript. 
1. The study's duration is only one year, and the sample size is minimal. The sample size for diabetes was merely 16, with a higher percentage of males than females. A balanced gender ratio in the sample would ensure that the findings apply to IPMN-associated pancreatitis in the research. The manuscript could benefit from more detailed explanations of acute pancreatitis, as this would enhance clarity and understanding of the topic. 

Answer: Thank you very much for the comment. We fully agree with the reviewer that the follow-up is short and that the sample size is limited. However, considering the rarity of acute pancreatitis associated to BD-IPMN and the previously published case series, the present paper represents the largest series on this topic with 135 patients. To underline again the limited number of patients and the short follow-up, the following sentence has been added: “Finally, although this is the largest series in the literature on this topic, larger studies with longer follow-up are needed to confirm these results.”  

Moreover, we agree with the reviewer that few patients with diabetes were included and significant conclusions on diabetes and BD-IPMN associted pancreatitis are not possible. A possible explanation is that the mean age was relatively young (55 years) at the time of the first acute pancreatitis and that the follow-up time is limited. However, an acceptable gender balance was present in the global study population, considering that 74 patients were males (54.8%) and 60 were females (45.2%). 

Finally, to better define the diagnosis of pancreatitis, the following sentence was added in the Methods section: “At least one episode of AP based on the Atlanta classification: presence of two of the following three criteria: typical abdominal pain (pain consistent with acute pancreatitis (acute onset of a persistent, severe, epigastric pain often radiating to the back), serum amylase and/or lipase > 3 times the upper normal limit, high-definition imaging (contrast-enhanced CT scan and/or contrast-enhanced MRI scan) suggestive of pancreatitis.” 

Round 2

Reviewer 1 Report

Comments and Suggestions for Authors

Dear Editor

The authors response to questions raised. I have no more questions.

Best

Author Response

Thank you very much.